# Duration of Bisphosphonate Drug Holidays in Osteoporosis Patients: A Narrative Review of the Evidence and Considerations for Decision-Making

**DOI:** 10.3390/jcm10051140

**Published:** 2021-03-09

**Authors:** Kaleen N. Hayes, Elizabeth M. Winter, Suzanne M. Cadarette, Andrea M. Burden

**Affiliations:** 1Dalla Lana School of Public Health, University of Toronto, Toronto, ON M5T 1P8, Canada; k.hayes@mail.utoronto.ca (K.N.H.); s.cadarette@utoronto.ca (S.M.C.); 2Center for Bone Quality, Division of Endocrinology, Department of Internal Medicine, Leiden University Medical Center, 2333 ZA Leiden, The Netherlands; E.M.Winter@lumc.nl; 3Leslie Dan Faculty of Pharmacy, University of Toronto, Toronto, ON M5S 3M2, Canada; 4Eshelman School of Pharmacy, University of North Carolina, Chapel Hill, NC 27599, USA; 5Institute of Pharmaceutical Sciences, Department of Chemistry and Applied Biosciences, ETH Zurich, 8093 Zurich, Switzerland

**Keywords:** osteoporosis, bisphosphonates, drug holiday

## Abstract

Bisphosphonates are first-line therapy for osteoporosis, with alendronate, risedronate, and zoledronate as the main treatments used globally. After one year of therapy, bisphosphonates are retained in bone for extended periods with extended anti-fracture effects after discontinuation. Due to this continued fracture protection and the potential for rare adverse events associated with long-term use (atypical femoral fractures and osteonecrosis of the jaw), a drug holiday of two to three years is recommended for most patients after long-term bisphosphonate therapy. The recommendation for a drug holiday up to three years is derived primarily from extensions of pivotal trials with alendronate and zoledronate and select surrogate marker studies. However, certain factors may modify the duration of bisphosphonate effects on a drug holiday and warrant consideration when determining an appropriate time off-therapy. In this narrative review, we recall what is currently known about drug holidays and discuss what we believe to be the primary considerations and areas for future research regarding drug holiday duration: total bisphosphonate exposure, type of bisphosphonate used, bone mineral density and falls risk, and patient sex and body weight.

## 1. Introduction

Osteoporosis is a disease characterized by low bone mineral density (BMD) and increased fracture risk. The pathophysiology of osteoporosis results from an imbalance in the bone remodeling cycle governed by osteoclast cells that break down bone and osteoblast cells that rebuild it, yet it is a complex disease with driving factors and manifestations in many other body systems [1]. One in three women and one in five men globally will experience an osteoporotic fracture, resulting in pain, disability, loss of independence, and even death [2,3,4].

Oral bisphosphonates are considered first-line therapy for the treatment of osteoporosis in many jurisdictions. Alendronate and risedronate are the most commonly used oral bisphosphonates, with robust evidence demonstrating their benefits on vertebral and nonvertebral fracture prevention in women and men with primary and glucocorticoid-induced osteoporosis (GIOP) [5,6,7,8,9]. Ibandronate is used less frequently as most meta-analyses show therapy has limited effects on hip fracture risk [10,11]. Major second-line therapies for osteoporosis include denosumab, intravenous bisphosphonates (zoledronate and ibandronate) and anabolic therapies (abaloparatide, teriparatide, and romosozumab). Other treatment options are less commonly used and include the lower potency oral bisphosphonate etidronate, raloxifene, and hormone replacement therapies among women at the time of menopause.

Bisphosphonates have a strong affinity for the bone mineral hydroxyapatite and bind via calcium-chelating properties, positioning them directly near osteoclasts, their target of action [12]. As bisphosphonate dissociates from hydroxyapatite, it enters activated osteoclasts to induce cell apoptosis. Variability exists between bisphosphonates in the mechanisms by which they cause cell death, yet most inhibit the enzyme farnesyl pyrophosphate synthase (FPPS) and prevent essential protein synthesis [12], with nitrogen-containing bisphosphonates (alendronate, risedronate, ibandronate, and zoledronate) providing the most potent in vitro effects on FPPS [12]. As bisphosphonates bind strongly to bone minerals, they can accumulate and endure for years in bone and circulation after therapy discontinuation [12,13]. Thus, as bisphosphonate slowly dissociates from hydroxyapatite, it continues to suppress bone turnover even after discontinuation of therapy [14,15,16]. Extensions of randomized controlled trials have shown that for many patients, the risk of most types of fracture does not increase after discontinuation of long-term alendronate therapy [14].

Given these anti-fracture effects after stopping treatment, guidelines advise that patients at a low to moderate risk of fracture undergo a two- to three-year drug holiday from bisphosphonate treatment after three to five years of therapy [11,17,18]. Drug holidays also reduce medication burden, particularly for older adults, and therefore present an important potential deprescribing opportunity. Patients at high fracture risk are recommended to continue bisphosphonate therapy or switch to another osteoporosis therapy. Notably, drug holidays are discouraged with non-bisphosphonate osteoporosis treatments, particularly denosumab, due to short-lived anti-fracture effects after discontinuation [17,19,20], and rebound vertebral fracture risk [21,22,23]. A full review of the post-treatment effects of denosumab has been conducted [24], and current evidence for treatment sequencing for non-bisphosphonate osteoporosis therapies has been recently described in the Endocrine Society clinical practice guidelines [11].

Another important consideration related to drug holidays is that very rare but serious adverse effects have been linked to prolonged bisphosphonate use [7,17,18,25]. Osteonecrosis of the jaw has mainly been documented after high bisphosphonate dosages (i.e., cancer indications), but atypical femoral fractures (AFF) have been linked to bisphosphonate treatment for osteoporosis [7,17]. Rates of AFF appear to increase after three to five years of bisphosphonate therapy, with estimates ranging from an odds ratio of 2.7 for an AFF among those with more than five years of therapy compared to those with less than five years [25] to a relative risk of AFF 63.5 when comparing those with eight to ten years of therapy to those with less than one year [26]. The variation in estimates is likely due to heterogenous study populations and outcome measurement [7]; true incidence of AFF in patients with and without bisphosphonate exposure is unclear. Other potential risk factors for AFF include Asian ethnicity, diabetes mellitus, rheumatoid arthritis, use of glucocorticoids and proton pump inhibitors, and osteopenia (versus osteoporosis) [17,27,28].

Thus, the rationale for a drug holiday is to harness residual bisphosphonate anti-fracture effects while stopping continued accumulation of drug and potentially reducing adverse events. Although initiating a drug holiday following bisphosphonate therapy among patients at low to moderate fracture risk is generally accepted, many questions remain regarding the optimal duration. In this narrative review, we summarize what is currently known about bisphosphonate drug holidays and outline four considerations that may influence clinical decision-making and future research regarding the duration of a bisphosphonate drug holiday once initiated.

## 2. Bisphosphonate Drug Holiday: What Is Known

Our understanding of the impact of drug holidays has largely come from extensions of randomized clinical trials, namely the alendronate and zoledronate pivotal trial extension studies [14,29,30]. For example, the alendronate pivotal trial extension (the Fracture Intervention Trial Long-term Extension, or FLEX, trial) examined the anti-fracture benefits of continuing alendronate treatment for 10 years, compared to those stopping treatment after five years [14]. While continuing therapy reduced the risk of clinically recognized vertebral fractures (HR 0.45 [95% CI 0.24–0.85]), no difference was found in nonvertebral (NVF) or morphometric vertebral fracture risk between groups. The zoledronate extension trial (the HORIZON trial) considered fracture risk after stopping three years of therapy with zoledronate [30] and detected no major differences in NVF for up to three years of a drug holiday. NVF, especially hip fractures, cause the majority of osteoporosis long-term morbidity and mortality [2,3,31,32,33]. Thus, discontinuation after five years of alendronate or three years of IV bisphosphonate therapy may not significantly increase the risk of NVF fracture for most patients. Reduced vertebral fracture risk up to one year after stopping three years of risedronate therapy has been demonstrated [16]. Further research is important to consider groups at higher NVF fracture risk during drug holidays as identified in post hoc analyses [29].

Generalizability of data from these extension trials to real-world patients has limitations. For example, the FLEX trial included postmenopausal women, most of whom were Caucasian and had low to moderate fracture risk [14]. In the past decade, studies have examined the effects of drug holidays on fracture risk in real-world patients, with varying conclusions. Recently, the FLEX trial was emulated using target-trial cohort study of U.S. women and also found no difference in five-year hip fracture risk between patients continuing versus discontinuing oral bisphosphonate therapy after five years (risk difference 3.8 per 1000 people [95% CI −7.4 to 15 per 1000 people]) [34]. A similar study conducted using Danish healthcare data for postmenopausal women also found no difference in hip fracture (incidence rate ratio 1.04 [95% CI 0.75 to 1.45]), vertebral fracture, or any major osteoporotic fracture between women who stopped versus continued alendronate after five years [35].

Other studies have examined fracture risk after varying lengths of bisphosphonate treatment. In 2008, Curtis and colleagues found an increased hip fracture risk among women who stopped oral bisphosphonates after at least two years of therapy (≥66% adherence) compared to continued use (HR 1.2 [95% CI 1.1 to 1.3]) [36]. A later study by the same group examined a cohort of women exposed to oral or intravenous bisphosphonates for at least three years with ≥80% adherence and found similar results when examining time off-therapy longer than two years [37]. Conversely, Adams and colleagues found no increased risk of fracture associated with any length of drug holiday compared to continued use among those with three or more years of use with ≥50% adherence (HR 0.95 [95% CI 0.83 to 1.10]) [38]. When results from observational studies published before 2017 were pooled in a meta-analysis, no increased risk of hip fracture was found (HR 1.09 [95% CI 0.87 to 1.37]) [39]. However, observational studies comparing medication users to non-users are frequently affected by unmeasured confounding and healthy user bias [40], potentially explaining conflicting findings. Differences in results may also arise from heterogenous baseline fracture risk (e.g., differences in age, race/ethnicity), varying data availability (e.g., laboratory data like BMD), and different bisphosphonate adherence thresholds (i.e., varying adherence required prior to drug holiday) [37].

Despite these real-world studies, knowledge gaps remain, as more nuanced depictions of drug holiday effects in diverse populations are critical. Specifically, no studies have examined the fracture effects of different lengths of a drug holiday by drug or patient characteristics, according to recent systematic reviews by Fink and Marchand [7,41]. Yet fracture risk over time during a drug holiday will likely change based on patient and therapy factors that affect duration of protective effects and baseline fracture risk [7,41]. Next, we highlight and discuss the top four factors we believe most important to consider when determining appropriate duration of a drug holiday, including the total bisphosphonate exposure prior to a treatment holiday, bisphosphonate therapy used, changing risk-factors, and patient-level characteristics.

## 3. Bisphosphonate Drug Holiday Duration: Considerations

### 3.1. Factor 1—Total Bisphosphonate Exposure: Treatment Duration and Adherence

Bisphosphonates have cumulative exposure effects, providing sufficient osteoclast inhibition and fracture protection only after six to 12 months of persistent use [12,42,43]. Anti-fracture effects after treatment discontinuation may be influenced by cumulative exposure as well, as residual bisphosphonate in the bone during a drug holiday is insufficient to reduce fracture risk indefinitely [44]. Pharmacologic studies have demonstrated relationships between duration of bisphosphonate treatment and duration of effects after treatment is stopped; for example, after 10 years of alendronate therapy, the effective dose in the body after treatment discontinuation is only about one-fourth of the on-therapy dose, with corresponding incremental decreases in bone turnover effects over time [44].

Importantly, the threshold treatment length of three to five years recommended prior to a drug holiday is based on a priori defined durations of the FLEX and HORIZON trials that were driven by study power [17]. Real-world zoledronate use appears mostly in-line with these treatment duration recommendations, with most patients beginning a drug holiday after three years of intravenous bisphosphonate treatment [37], yet some studies indicate many patients undergo a drug holiday after fewer than five years of oral bisphosphonate treatment [37,45,46]. In addition, in clinical practice, low adherence has been frequently documented with oral bisphosphonate treatment [47,48,49] and so some patients may have suboptimal adherence prior to a drug holiday.

Whether fracture risk is increased during a drug holiday after lower total bisphosphonate exposure (i.e., shorter durations of or lower adherence to therapy) is unknown. Yet based on the evidence to date, we would expect that shorter periods of therapy (e.g., less than 3 years for IV bisphosphonate or five years for oral bisphosphonate therapy), or low adherence to therapy [36] likely reduce the length of continued fracture protection during a drug holiday. For example, a drug holiday of one year may be more appropriate in patients with less total bisphosphonate exposure (e.g., three years of oral bisphosphonate treatment at low adherence, similar to 1.5 years of total exposure) due to lower residual bisphosphonate protection, though robust studies examining the effects of adherence and total cumulative bisphosphonate exposure are lacking.

### 3.2. Factor 2—Bisphosphonate Used Prior to Drug Holiday

Bisphosphonates not only have differential potency of osteoclast inhibition, they also have varying affinity for hydroxyapatite and are cleared from the body at differential rates [12], which may impact the choice of drug holiday duration. Table 1 provides an overview of the pharmacologic properties of the alendronate, risedronate, and zoledronate, and the current recommendations for the duration of a drug holiday.

Zoledronate has the greatest affinity for hydroxyapatite, with the following bisphosphonates having decreasing affinity: alendronate, ibandronate, risedronate, and etidronate [12,18]. In comparison to alendronate and zoledronate, risedronate has a weaker affinity for hydroxyapatite and also a less positively charged molecule that may prevent accumulation of bisphosphonate in bone [12]. As risedronate might accumulate less and be eliminated more quickly from the body, fracture protection may be more fleeting after therapy is stopped. While the FLEX and HORIZON trials demonstrated that alendronate and zoledronate continue to suppress bone turnover markers (BTM) for up to five years and three years after therapy discontinuation, respectively [14,30], BTM returned to placebo levels within one year after risedronate was stopped [16], potentially justifying shorter drug holiday lengths. A recent study did not identify differences in BTM changes at 96 weeks after discontinuation of alendronate versus risedronate therapy [50], yet a secondary analysis of an observational drug holiday study suggests that risedronate treatment may be associated with higher fracture risk than alendronate after long-term therapy is stopped [37]. At minimum, patients on a risedronate drug holiday may need to be more carefully monitored for fracture risk factor changes during a drug holiday than those discontinuing alendronate therapy; a drug holiday of one to two years after risedronate therapy may be most appropriate [17].

### 3.3. Factor 3—Risk Factors: Bone Mineral Density and Fall Risk

Changes in risk factors for fracture are perhaps the most important consideration when re-assessing appropriateness of a continued drug holiday. Risk factors increase the probability of fracture without osteoporosis therapy, and the risk-benefit ratio of a drug holiday diminishes. For example, the rate of atypical fracture after 8 years of bisphosphonate therapy is 78 per 100,000 person-years [18], while the rate of major osteoporotic fracture among untreated women with moderate fracture risk is 1600 per 100,000 patient-years [18,53]. Given the strong evidence and magnitude of therapeutic benefit versus the weaker evidence of drug holidays to reduce adverse events, a change in a major fracture risk factor therefore may provide a compelling rationale to cease a drug holiday and re-initiate osteoporosis therapy [18].

Fracture risk assessment tools, like FRAX [54], have not been validated in patients undergoing a drug holiday and so have less clinical utility in this context. Examination of specific fracture risk factors may therefore be most helpful when determining duration of a drug holiday. Although all fracture risk factors are important to consider when initiating the drug holiday (e.g., age, sex, prior fracture, continuation of glucocorticoid use [53,55,56]), here we focus on factors that are most likely to change during a drug holiday. These time-varying risk factors include bone mineral density (including initiation of medications that may lower BMD) and fall risk (including initiation of medications that may increase fall risk) [55,57]. In general, we lack validated predictive markers of fracture risk for treated patients, yet biochemical markers currently being explored to predict fracture risk that may be extrapolated to those on a drug holiday include high-resolution peripheral quantitative computerized tomography, trabecular bone scores, and impact microindentation [58].

Though not a precise predictor of fracture risk [59], decreasing BMD values are associated with a higher risk of fracture [60]. Evidence is varied regarding the benefits of BMD monitoring while on a drug holiday [17,61,62], and fracture risk appears to be more independent of surrogate marker changes after therapy discontinuation than while on treatment [16]. However, changes in factors known to affect fracture risk and/or BMD might still provide evidence that informs whether an ongoing drug holiday is safe. Medications with the strongest evidence for effects on BMD and subsequent increased fracture risk include chronic glucocorticoid therapy and aromatase inhibitors (e.g., anastrozole, letrozole, exemestane) [55,57]. Other medications that may affect BMD but potentially do not justify ending a drug holiday include androgen deprivation therapy and some chemotherapeutic agents, thiazolidinediones, proton pump inhibitors, sodium-glucose co-transporter-2 inhibitors, chronic heparin therapy, and thyroid hormone supplementation [57]. Clinical judgement and consideration of other fracture risk factors is important to determine whether a drug holiday remains appropriate if these medications are initiated.

BTM may be an important monitoring parameter during drug holidays to determine if bone strength will be maintained after therapy discontinuation. BTM increases are predictive of BMD losses during a drug holiday [63,64]. For example, among men and women from one hospital in Denmark, an increase in plasma C-terminal peptide (p-CTX) within three months after discontinuation alendronate therapy (median treatment duration 7.0 years [range 5.0–20.0 years]) was predictive of greater BMD declines at the total hip site one year later [63]. In addition, variability exists in the degree of BTM changes after long-term bisphosphonate therapy discontinuation, indicating that BTM may distinguish between patients who may or may not benefit from re-start of therapy after a drug holiday. Among 158 patients from a UK outpatient specialist center who initiated a drug holiday after at least five years of bisphosphonate therapy (17% men, 69% alendronate), 25% had significant increases in p-CTX in the first four months off therapy, while half experienced no detectable changes in p-CTX up to 1 year later [65]. Whether varying patterns of BTM changes are able to predict fracture during time off therapy is unknown. More research is needed to inform BTM monitoring recommendations on drug holidays.

Changes to fall risk are also an important fracture risk factor to consider throughout a drug holiday [18]. While there are a number of factors that may increase fall risk, changes in medication use can be straightforwardly monitored. The Beer’s Criteria for Potentially Inappropriate Medication Use in Older Adults provides a comprehensive list of medications that increase fall risk (e.g., benzodiazepine therapy) [66]. Avoidance of medications that increase fall risk is the ideal strategy for older adults and most patients with osteoporosis [66], yet occasionally the benefit of these medications outweighs this risk (e.g., anticonvulsant or antihypertensive medications). Unfortunately, no available evidence examines the impact of new fall risk factors during a drug holiday, or determines if re-initiation of bone-strengthening medications like bisphosphonates prevent fall-induced fractures after a drug holiday. Nevertheless, if a patient’s fall risk increases, it may be appropriate to end a drug holiday to mitigate fall-related fracture risk in addition to implementing fall reduction measures [67].

### 3.4. Factor 4—Patient Characteristics: Sex and Body Weight

Robust evidence shows that fracture risk in treatment-naïve and treated people with osteoporosis is modified by characteristics like female sex [68], low body weight [69], and Caucasian ethnicity [70,71]. However, to date, no study has examined how drug holiday effects could be impacted by these characteristics [7]. Careful consideration of potentially differential effects of drug holidays by sex (factors that are biological) and gender (cultural and social influences based on sex) [72] is paramount. Bisphosphonates were developed when women were essentially the exclusive recipients of osteoporosis diagnoses and treatments [73]. Misunderstandings of the risk of osteoporosis in men remain problematic among both patients and clinicians [74], and osteoporosis remains more underdiagnosed and undertreated in men than women [68,75,76].

The landscape of osteoporosis treatment in men has changed drastically in the prior two decades. Data show that although women do experience primary osteoporosis at higher rates than men due to the decline in estrogen from menopause, men also experience biological changes in aging that make them susceptible to osteoporosis [68]; a reduction in testosterone can contribute to BMD decline in up to 1 in 4 older men [77,78]. In addition, up to 50% of osteoporosis cases in men are driven by secondary causes (e.g., GIOP or osteoporosis secondary to hypogonadism [e.g., from androgen deprivation therapy]) [68,77,79,80] and men are living longer today with more age-related comorbidities and frailty than when bisphosphonates came to market [81]. New understandings of these risks of osteoporosis in men have led to significant advancements in treatment. In Canada, for example, almost ten-times more men are initiating bisphosphonate treatment today versus 20 years ago [46,82].

Despite the higher prevalence of osteoporosis medication use in men, however, no large empiric studies have examined the impacts of long-term bisphosphonate therapy or drug holidays in men. Landmark trial extensions and current observational evidence that provide the bulk of evidence on drug holidays of two to three years’ duration have been conducted exclusively in women [7,15,42,43]. However, there are important musculoskeletal and healthcare delivery differences between men and women that likely modify drug holiday effects [68]. First, men are more likely to have secondary osteoporosis. In this case, monitoring changes in the cause of osteoporosis may help to guide both the decision to start a drug holiday and also determine the optimal duration once initiated [83]. For example, in males with hypogonadism-induced osteoporosis who are being treated with testosterone and otherwise at low fracture risk, a bisphosphonate drug holiday may be appropriate. Similarly, patients with GIOP whose BMD returns above osteoporotic levels after glucocorticoid discontinuation may be able to remain off bisphosphonates unless glucocorticoid therapy is re-initiated. However, continual monitoring of these patients is required, as a subsequent BMD decrease may indicate to the clinician that bisphosphonate therapy should be re-initiated. To date, little data examine the effects of osteoporosis therapy effects in men, let alone the effects of drug holidays [77].

Next, musculoskeletal composition differs between men and women and may drive changes in bisphosphonate therapy effects. Men have higher muscle mass than women that may help to ameliorate fall impact; however, a man is 50% more likely to die after experiencing a hip fracture than a woman with similar risk factors [3,68,84]. Importantly, the diameter of men’s bones is larger than those of women of the same height and weight, and the average man’s body surface area is 1.5-times greater versus the typical woman [79]. Men may therefore have lower total bisphosphonate exposure than women even with the same dose and duration of treatment. In addition, larger bone structure in men may also cause AFF to manifest differently than in women [68], yet studies and reports of AFF are almost exclusively in women [85]. ONJ risk appears to be similar between men and women [86]. Finally, osteoporosis medication nonadherence is twice as likely among men than women [68], contributing to further lowered levels of bisphosphonate exposure that may affect fracture risk during a drug holiday.

Body weight and composition may also modify the duration of bisphosphonate effects during a drug holiday independently of sex. Unlike other therapies that are dosed based on body surface area to maximize effectiveness while mitigating adverse effects [87], like anti-cancer agents, patients with larger body size receive the same dosage of bisphosphonate therapy as those with smaller body frames or weight. Bisphosphonates concentrate in areas of the bone with higher remodeling rates (e.g., lumbar and thoracic vertebrae) but are released from bone into circulation after therapy discontinuation to aid in bone remodeling correction at other sites (e.g., femoral neck) [12]. Thus, even with all other fracture risk and prior therapy factors being equal (i.e., treatment is indicated and the patient has low BMD), people with larger body frames may not derive the same lasting anti-fracture effects during a drug holiday as those with smaller frames due to less remaining bisphosphonate in proportion to bone size. More studies on the differential fracture risk by patient characteristics is needed to conclude if sex-specific or body-weight specific drug holiday timeframes should be implemented.

Race and ethnicity may also play a role in the effects of drug holidays and appropriate durations; however, studies have not elucidated whether these complex effects are independent of body size, musculoskeletal composition, and societal factors (i.e., racial disparities) [71]. Nevertheless, evidence indicates there is fracture risk modification by race and ethnicity in women. Cohort studies have shown that in the US, Caucasian women experience the highest rates of fracture [88] and Black women, despite having more muscle mass on average, experience the most severe fracture sequelae [70]. Moreover, adverse effects associated with excessive accumulation of bisphosphonate, like AFF, appear more common among women of Asian descent [17,28]. Little data exist on ethnic and racial differences and disparities in men with osteoporosis. All other factors considered, based on the current evidence, a longer drug holiday may be more appropriate for patients of Asian ethnicity versus Caucasian or Black patients, due to lower major osteoporotic fracture risk and higher risk of adverse events from long-term bisphosphonate therapy. However, more evidence is needed to directly compare drug holiday effects by ethnicity and to disentangle the causal relationships between body composition, ethnicity, race, sex, and fracture.

## 4. Conclusions

Oral bisphosphonates have contributed to the decline of osteoporotic fractures globally. Strong evidence supports drug holidays for most patients after three to five years of continuous bisphosphonate therapy, yet there is a lack of evidence informing lengths of drug holidays. The appropriate duration of a drug holiday is likely affected by the initial length and concentration of therapy, drug retention pharmacology, BMD and fall risk, and patient characteristics, like skeletal size. Future research is needed to examine fracture risk by drug holiday length that considers how these factors modify fracture risk to form patient- and therapy-specific drug holiday timeframes.

## Figures and Tables

**Table 1 jcm-10-01140-t001:** Pharmacologic Properties and Current Drug Holiday Duration Recommendations for the Most Commonly Used Bisphosphonates.

Bisphosphonate	FPPS Inhibitory Potency [12,51]	Binding Affinity to Hydroxyapatite [12,52]	Accumulation and Saturation Potential [12,52]	Current Recommended Drug Holiday Duration [17]
Alendronate	Low	High	High	2–3 years
Risedronate	High	Low	Low	1–2 years
Zoledronate	High	High	Low	2–3 years

## Data Availability

No new data were created or analyzed in this study. Data sharing is not applicable to this article.

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
