# Peer review of "Duration of Bisphosphonate Drug Holidays in Osteoporosis Patients: A Narrative Review of the Evidence and Considerations for Decision-Making"

_jcm, 2021, doi:10.3390/jcm10051140_

Round 1

Reviewer 1 Report

This is a well written paper about the duration of bisphosphonate drug holidays in osteoporosis cases.

Bisphosphonates can be used to reduce the risk of  osteoporosis. However, several aspects of this treatment remain unclear.

Further studies about the duration of drug holiday are essential, thus I believe that this kind of studies are valuable contributions to scientific society. 

Considerations are interestingly presented. 

Reviewer 2 Report

You have failed to include any analysis of the use of bone turnover markers and a drug holiday. For example:

 Statham L, Abdy S, Aspray TJ. Can bone turnover markers help to define the suitability and duration of bisphosphonate drug holidays? Drugs in Context 2020; 9: 2020-1-3. DOI: 10.7573/dic.2020-1-3

Bone turnover markers are a  clinical tool and a discussion  should be included.

Reviewer 3 Report

The diagnosis and management of osteoporosis has gained tremendous interests among several medical disciplines.  Antiresorptive agents have been used and considered the most effective therapeutic advances to combat fractures and to correct and rebuild skeletal losses. The latter statement is on one hand partly true, but on the other hand we cannot control optimal rebuilding processes of skeletal losses in osteoporotic patients. Simply because osteoporosis is not a diagnostic entity, it´s mostly a symptom complex. In other words, bone fragility is part of a multisytem disease, Focussing on one pathological ailment in osteoporotic patients is not a wise approach for proper management. BMD results and the other risk factors of the FRAX algorithm are just co-factors and in genuine practice are not of diagnostic importance. Every patient shuold be documenetd upon his/her individualistic bases (clinical and radiological phenotypic characterizations). Throwing all osteoporotic patients in one basket is  a great mistake.Authors need to refer to London dysmorpholgy database you will find hundreds of reasons behind fractures(fractures are just part of a multi-system involvement). The vast majority of colleagues work in the management of osteoporosis are endocrinologists. Their expertise in bone malformation complex and bone disorders in general is very modest. The main concept of  this paper is the “biphosphonate holiday”, which has been applied widely and for long time and it was and still as a subject of contoversey. One question imposes itself ; Is it sufficient for Post-holiday patients to be  re-assessed only via  the FRAX or any similar tools of the task force ?  Are these tools sufficient to decide? Scientifically and from the biological respectives; With Ageing  all chemical  compounds are accumulated in the lumbar and pelvic region, on the one hand it lessens the fracture rate, but on the other hand it turns this region into a depot of toxins.

Round 2

Reviewer 2 Report

updates appear to be appropriate

Reviewer 3 Report

The changes are persuasive
